# An Empirical Investigation of Environmental Knowledge and Attitudes as the Correlates of Environmental Identity among Pre-Service Biology Teachers in Tanzania

Josephat Paul Nkaizirwa [1,2,*], Catherine Musalagani Aurah [3] and Florien Nsanganwimana [1]

1   African Centre of Excellence for Innovative Teaching and Learning Mathematics and Science (ACEITLMS), College of Education, University of Rwanda, Kigali P.O. Box 4285, Rwanda
2   Department of Educational Psychology and Curriculum Studies, College of Education, The University of Dodoma, Dodoma P.O. Box 523, Tanzania
3   Department of Science and Mathematics Education, School of Education, Masinde Muliro University of Science and Technology, Kakamega P.O. Box 190-50100, Kenya
*   Correspondence: pauljosephat@yahoo.com or josephat.nkaizirwa@udom.ac.tz; Tel.: +255-679-262-614

**Abstract:** Despite the extant literature that discusses the potentiality of environmental identity in shaping people's engagement with nature, there is little evidence of which specific domains of environmental attitudes and knowledge are connected with environmental identity. The present article reports on the results from two studies that were conducted within the framework of the Solomon four-group design, in four randomly selected teacher colleges (TCs) in Tanzania. Specifically, study one was an intervention with pre- and post-measurements that involved indoor and outdoor environmental programs among pre-service biology teachers in two TCs. Moreover, study two was conducted in two other selected TCs, with a post-test only. The two-Major Environmental Values (2-MEV) and a Competence Model for Environmental Education were used for measuring environmental attitudes and knowledge, respectively. Finally, environmental identity was measured using the environmental identity (EID) scale. The results revealed that preservation was positively correlated with domains of identity, while other variables demonstrated overlapping relationships in different measurement points. Moreover, notable correlations between social desirability responding with some domains of attitudes and identity were evident, whereas the age of respondents had limited associations with other variables. The article concludes by proposing the need to promote environmental attitudes (EA) and knowledge as important strategies in fostering environmental stewardship.

**Keywords:** competence model for environmental education; environmental attitudes; environmental identity; the environmental identity (EID) scale; environmental knowledge; pre-service biology teachers; two-major environmental values (2-MEV)

## 1. Introduction

Previous research has reported that there is remarkable evidence indicating the usefulness of environmental identity in fostering environmental stewardship [1,2]. Operationally, environmental identity (EI) has been described as the extent to which people associate themselves with the natural environment [3]. Meanwhile, the level of association that individuals attach to the natural environment varies considerably between and among societies of different backgrounds. Particularly, evidence in the literature suggests that rural–urban residents tend to possess different levels of their relationships with the environment. Likewise, people with different psychological constructs, such as attitudes, emotional well-being, and personal meaning, among others, tend to demonstrate different degrees of association with nature. Despite the usefulness of EI in shaping people's relatedness with nature, little effort has been dedicated to understanding the specific factors of environmental attitudes and knowledge that connect with specific domains of environmental

identity in human–nature relations, not only in low- and middle-income regions such as sub-Saharan Africa, but also on a global scale [4].

International efforts throughout the United Nations (UN) have called upon the need to protect the environment, as all member countries strive to achieve the UN sustainable development goals (SDGs) [5]. However, rapid population growth has increased inequality and pressure on natural resources, resulting in increased loss of biodiversity, climate change, and environmental pollution, among other environmental challenges [6]. Comparatively, Africa is expected to suffer the most detrimental consequences of environmental challenges [7,8]. Consequently, this calls for an agency with a transformative approach to changing how people orient themselves to nature.

Given the UN SDGs, the themes that focus on attaining environmental sustainability are directly connected to society, and they should be reflected in people's environmental knowledge, attitudes, and commitments [9]. Accordingly, understanding the factors that underscore the extent to which people associate themselves with nature is an imperative step to fostering environmental stewardship [10]. Nevertheless, limited effort has been dedicated to determining the underlying factors explaining EI, particularly in developing countries such as Tanzania. As a result of inadequate empirical support from limited studies, implementing environmental policies and related frameworks have become somewhat haphazard [11]. Thus, the need to undertake environmental research is paramount, with a particular focus on revealing people's orientation to nature and how their nature relations change with other psychological and cognitive aspects. Predominantly, understanding EI is crucial, given that there is considerable emphasis on other environmental constructs in previous research [12]. In order to attain this purpose, the following research questions were systematically investigated, using two studies reported in this study:

1.  How many domains of environmental identity can be explained by environmental attitudes and environmental knowledge dimensions?
2.  Which domains of environmental attitudes and knowledge have the strongest significant correlation with specific domains of environmental identity?
3.  Given the study interventions in different study contexts, can the dimensions of environmental attitudes and knowledge retain the same variability of explained relationships with the domains of environmental identity?
4.  Do respondents' ages and social desirability responding pose considerable impacts on the relationships between environmental knowledge dimensions, attitudes, and specific domains of environmental identity?

We believe that the results presented in this study provide a novel contribution to understanding environmental identity and its relationships with environmental attitudes and knowledge. In turn, the fundamental research designs and future policy reforms can be deduced on the basis of the outcomes of these results. Ultimately, environmental educators are provided with areas of concern that need to be strengthened in fostering environmental actions in learning institutions in Tanzania, and possibly beyond.

## 1.1. The Context of the Environmental Identity Scale

Developmentally, the first attempt to develop an environmental identity (EID) scale was conducted using United States (U.S.) college students, from which 24 items were published as reliable and valid measures of environmental identity [13]. Subsequently, other independent studies in different study contexts were conducted, in order to validate the EID scale [14,15]. While the original development of the scale proposed a unidimensional construct of the EID scale, with some support from independent researchers, exceptions to the model have also been reported [16]. Due to limited multicultural representation in the original scale, a revised version was developed and tested in other five multicultural countries from four continents [17]. The revision of the original EID scale resulted in 14 items, with a modification of 2 additional items that included emotional connectedness with nature. Nevertheless, the revised EID scale that included samples from the U.S., Peru, Taiwan, Switzerland, and Russia, did not include any samples from the African continent.

This perpetuates the existing gap in sub-Saharan African countries, which are generally underrepresented in the majority of environmental psychology studies [18].

One recent validation of the revised EID scale found that two constructs were positively correlated with a pro-environmental identity [19]. On the other hand, a qualitative assessment to explore how EI develops among undergraduate students at Duke University was conducted by Miao and Cagle [20]. It reported different influential factors, specifically that outdoor exposure, such as vacation and traveling, formed the most significant contribution, followed by the habitat within which an individual spends most of his/her time, formal education, and peer groups. Furthermore, parents were recognized as being the most influential factor at the family level (70%), whereas teachers/mentors (37%) and elders (30%), apart from parents, were among the potential determinants.

Another study that was conducted by Walsh and Cordero [21] in the U.S. integrated digital content of filmmaking about climate change and science content development among youths. They found that the integration of digital content, such as storytelling, helps to improve youths' EI. In addition, Simms and Shanahan [22] investigated how EI develops within the complex social structures of classroom contexts in Canada, by exposing participants to marine pollution. They observed that the program played a pivotal role in shaping inquiry, emotional, and critical reflections on the participants' environmental identity. Additionally, Simms [23] reviewed the extant literature, in order to assess how EI is currently being operationalized and researched. She found that Clayton's interpretation of EI, which focuses on the physical aspect, has influenced contemporary research on EI, despite the existence of two other interpretations in the education context: Eriksonian identity theories and Meadian identity theories. Broadly, evidence in the literature suggests that further research is necessary, in order to better understand EI. More importantly, the African context research is prudent, given the limited focus of such studies in the region. Our results are among the few of the primary discussions that are necessary to initiate subsequent debates in the African region and beyond.

### 1.2. Environmental Attitudes

The measurement of EA can be traced back to the late 1960s and early 1970s, when the primary focus was to assess the level of support for environmental concern, and the acceptance of environmental policies in most European countries [24]. It has been argued that most environmental problems are associated with individual beliefs, attitudes, and behavior, and that their redress cannot be separated from the source that created them. In support of the Dominant Social Paradigm (DSP), Kilbourne et al. [25] argued that when one's concern for the environment increases, the willingness to protect and conserve the environment also increases. This idea was extended by Dunlap and Liere [26], who proposed the New Environmental Paradigm (NEP) scale. The development of the NEP was initiated by the need to bring ecological balance, while eliminating the anti-ecological perspective, of utilizing nature for human benefits. Based on the NEP scale, an individual can either possess an ecological perspective, or an anti-DSP (anti-ecological) view, but not both. Later in 2000, the NEP scale was revised to capture more inclusive aspects of pro- and anti-NEPs, as well as avoid automated terminologies and gender-biased wordings of items [27].

The revision of the NEP scale was also motivated by increasing trends in complex environmental problems that were geographically dispersed and complex to measure [27]. Despite being the most widely used scale of EA, the NEP has been criticized for being unidimensional in nature, which forces individuals to be placed on one environmental dimension [28]. Another useful measure of EA was presented by Kaiser et al. [29], who systematically developed five scales of EA, and found that they were measuring the same attributes. Ultimately, they mainly focused on the Campbell Paradigm in operationalizing EA. Nonetheless, unlike other measures, the Campbell Paradigm requires application of the Rasch model in the measurement of EA. Broadly, three measures of EA have become popular, despite a considerable number of others. The three popular scales are the En-

vironmental Concern Scale [30], the Ecology Scale [31], and the NEP scale [27]. Yet, the majority of the items focused on measuring EA as a horizontal structure (unidimensional), with only a few having investigated the vertical structure or multidimensional constructs of EA [32,33].

The existence of previous research underlying EA as either unidimensional or multidimensional entailed that EA is formed by primary or first-order factors, and secondary or higher-order factors. However, none of the existing scales attempted to measure EA in its entirety [34]. Consequently, the development of the Environmental Attitudes Inventory (EAI) by Milfont and Duckitt aimed to address this challenge. Thus, two versions of the EAI were proposed, the short version scale (EAI-S) composed of 72 items, and a brief version with 24 items (EAI-24). Nonetheless, the EAI has been critiqued for being too long and taking long to complete, while optimized items by subsequent studies resulted in easier completion of the items [35–37]. Ultimately, the two-major environmental values (2-MEV) measuring EA on the two-factor structure of preservation and utilization have become popular. Specifically, preservation measures the protection and conservation of nature (ecocentric view), whereas utilization measures the preference of individuals in using environmental resources (anthropocentric perspective). The 2-MEV presents two major advantages in measuring EA [38]. First, it allows an individual to be placed on either of the two dimensions without necessarily affecting each other. Second, the scale is short and easy to complete. The 2-MEV has been widely validated, and has been produced in more than 30 different language versions [39].

### 1.3. Environmental Knowledge

Mediating environmentally friendly actions requires an interplay between attitudes, knowledge, and behavior [40]. In order to achieve this tripartite relation, the Competence Model for Environmental (CMEE) education provides a robust understanding of these three factors. The CMEE was initially developed by Kaiser et al. [41], in order to measure environmental knowledge in three dimensions. Particularly, the environmental knowledge dimensions that are measured in CMEE are system knowledge (knowledge about the relationships between organisms in an ecosystem), action-related knowledge (knowledge about how to achieve environmental protection and conservation), and effectiveness knowledge (knowledge about how to best achieve nature preservation) [42]. The CMEE has been empirically validated, and its psychometric properties are remarkable [43]. Past research has provided considerable evidence on the relationships between knowledge, EA, and behaviors [44–46]. However, mixed results have also been reported, which suggested that preservation and knowledge dimensions were not correlated [47]. The ceiling effect was partly attributed to be one of the reasons for this observation. Likewise, Paço and Lavrador [48] did not find a relationship between environmental knowledge and EA. Given these contrasting results, further research is important.

In general, considerable past research has, to a great extent, focused on understanding the relationships between knowledge, EA, and behaviors. Given the usefulness of environmental identity in shaping pro-environmental behaviors, we argue for the need to re-examine the correlates of environmental identity. Primarily, past research has established the distal relationship between knowledge and behavior; however, little of this is knowledge regarding the extent to which domains of environmental identity depend on attitudes and knowledge in their formation, particularly in the education context. Thus, our primary focus was to provide a novel contribution to the usefulness of EA and knowledge in explaining their relationships with specific domains of environmental identity. The results help to shape future discussions, not only in curriculum development and implementation, but also on the specific emphasis that is necessary to be integrated into environmental and education policies, as well as in their implementation frameworks, in contexts similar to ours.

## 2. Materials and Methods

### 2.1. Participants, Study 1

Study one engaged 160 (53.8% females) pre-service biology teachers from two randomly selected TCs that were located in two different regions in Tanzania, named hereafter $TC_1$ and $TC_2$, respectively. The selected TCs offer diplomas in the secondary education program, and selected participants were in the final year of their academic studies. This was considered, given that they had remarkable experience having stayed in TCs for at least two years, making them good candidates to provide reliable data. The age (in years) of subjects in $TC_1$ ranged from 17 to 28 years (females M = 20, SD = 2.1; males M = 21.4, SD = 2.6), whereas in $TC_2$ the age of subjects ranged from 18 to 30 years (females M = 20.7, SD = 1.4; males M = 22.7, SD = 2.7). Specifically, 91 subjects (males = 53.8%) were obtained in $TC_1$, while only 69 (females = 63.8%) subjects were used for the same study in $TC_2$. The number of subjects in each TC was determined by the number of pre-service biology teachers in the respective year of academic study that was considered for the purpose of the present study.

### 2.2. Participants, Study 2

Study two involved 173 (50.9% males) pre-service biology teachers, again from two randomly selected TCs named hereafter as $TC_3$ ($n$ = 95; females = 53.7%) and $TC_4$ ($n$ = 78; males = 56.4%), respectively. The age of the subjects in $TC_3$ ranged from 19 to 32 years (females M = 21.7, SD = 1.9; males M = 23.1, SD = 3.0), whereas in $TC_4$, subjects' ages ranged from 17 to 30 years (females M = 19.4, SD = 1.8; males M = 20.6, SD = 2.7). In general, the participants in all of the groups were approximately equivalent in terms of age and gender, and they were pursuing similar diploma programs in the same year of academic study as those in study 1. Taken together (studies 1 and 2), the study involved 333 pre-service biology teachers (171 females; 162 males). Nonetheless, each sample was treated as an independent sample in both studies, and there was no effort to combine more than one sample into a single unit of analysis in both studies.

### 2.3. Sampling Procedures

Since the study was conducted in science-offering TCs in Tanzania, the initial stage began by identifying the target population that was enrolled in science-offering TCs in Tanzania. Particularly, 10 TCs were identified as the only public TCs that offered science-related diploma programs for secondary education. Therefore, after identifying 10 TCs as the potential clusters for the study, 4 TCs out of 10 from the recognized list of TCs were simple randomly selected for the study [49]. Precisely, a single-stage cluster random sampling was employed, in order to obtain four TCs for the study [50]. For confidentiality, the selected TCs were assigned codes from one to four ($TC_1$, $TC_2$, $TC_3$, $TC_4$), as stated earlier. Given the formula proposed by Tabachnick and Fidell [51], N > 50 + 8 m (where m = number of independent variables), a minimum sample size of 90 was required to assess the predictive power of independent variables over outcome variables of environmental identity, as five predictors were employed in the study. However, combining different samples into a single study group and treating it as an independent sample leads to a sacrificial pseudoreplication [52,53]. Therefore, to avoid an inflated degree of freedom and the possibility of committing type II errors in the interpretation of the study findings, each sample was treated separately during the analysis and presentation of the study findings [54].

For ethical considerations, participants' consent was requested before they became actively engaged in the study. Moreover, the purpose of the study was communicated in advance to the participants, prior to their engagement in the study's activities. Figure 1 provides details about the sample size.

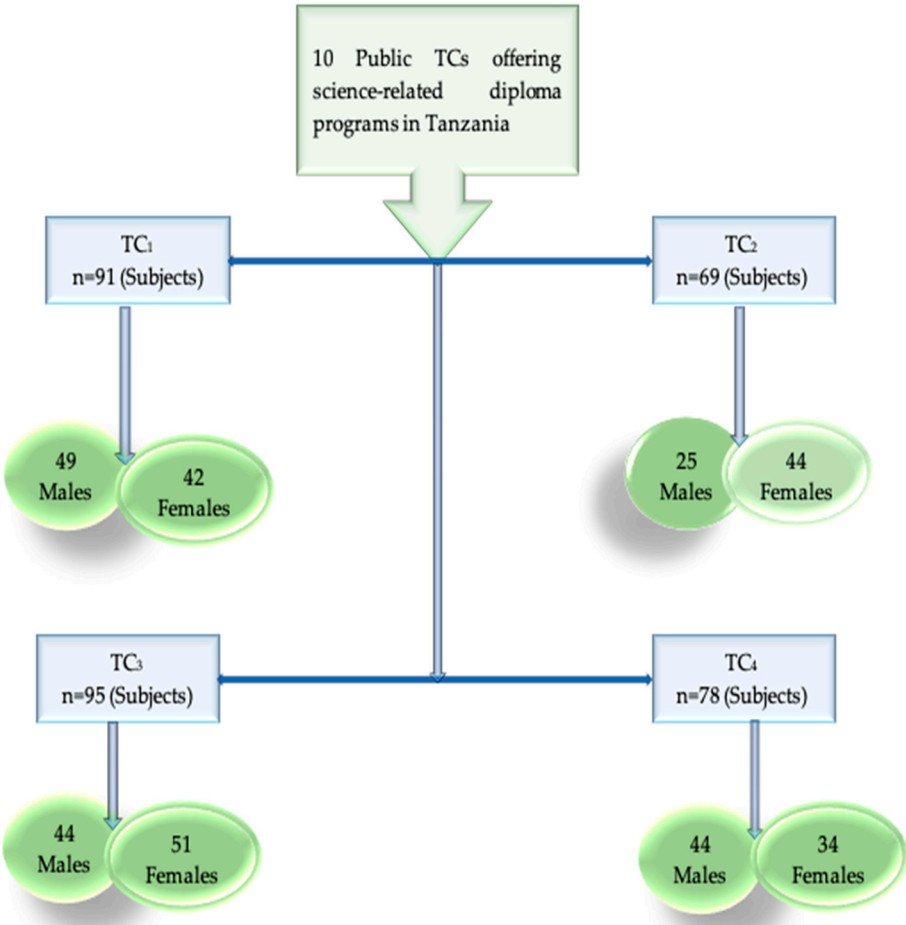

**Figure 1.** A summarized flowchart of the study samples.

*2.4. The Study Interventions*

The primary aim of the study was to improve environmental knowledge and attitudes through which environmental identity was systematically observed, in order to document how it changes as the attitudes and knowledge change throughout the study interventions. Therefore, the content of the study intervention was designed on the basis of two themes of the teacher education curriculum, particularly the biology academic syllabus for diploma programs for secondary education. Specifically, two themes were modified and delivered to pre-service biology teachers. The first theme was "the effect of rapid human population growth on the environment", and the second theme focused on the "effects of pollution on living things and the environment" [55].

Using differentiated methods, the identified themes were interactively discussed with pre-service biology teachers. For the study groups that were identified as targets of the experimental treatment through the random assignment procedure, an intervention was conducted using the inquiry-based learning (IBL) strategy, whereas other study groups continued with conventional approaches that are traditionally used in TCs. Situated in Experiential Learning Theory (ELT) [56], the intervention was conducted using indoor and outdoor learning activities. Outdoor learning activities involved physical observations of environmental learning aspects by pre-service biology teachers within the selected teacher college campuses. The physical observations included waste management strategic areas, methods used for laboratory chemical spills in TCs, a site visit of areas affected by increased human population, as well as gardening activities within the TCs. Indoor learning activities involved reflective discussions on the key issues that were observed during the outdoor learning activities, report analyses of participants' groups, and proposing modifications for subsequent learning activities. A maximum of three weeks was used to administer the

program in each of the selected intervention TCs. The study was conducted from January to April 2021.

### 2.5. Data Collection Instruments

Data collection was carried out using three kinds of research instruments. For measuring environmental attitudes, 20 items of the 2-MEV were employed, out of which 10 items measured preservation (PRE) of nature, and 10 others measured utilization of nature (UTL) [38]. The 2-MEV was scored on a five-point scale, from 1—strongly disagree to 5—strongly agree. Furthermore, 19 items of the EID scale were used for measuring the environmental identity [13]. The EID scale was rated by the participants using a seven-point scale, from 1—not at all true of me to 7—completely true of me. In addition, 21 items were used to assess environmental knowledge dimensions of the system knowledge, action-related knowledge, and effectiveness knowledge. Four alternatives were used in each item of the environment knowledge test (EKT). Finally, all the variables of interest were controlled for social desirability responding (SDR), using the 16 items of the Balanced Inventory of Desirable Responding Short Form (BIDR) [57]. Participants completed the BIDR-16 on a five-point scale, from 1—strongly disagree to 5—strongly agree. The Environmental Knowledge Test (EKT) consisted of multiple-selection items, as well as fill-in and supply items, that required the participants to provide a more comprehensive understanding of the measured cognitive aspect. Permission to use the adapted scales was requested and granted by the respective authors.

### 2.6. Pilot Study

A pilot study was conducted among the 76 pre-service biology teachers from a different TC that did not take part in the main studies. Participants' ages ranged from 19 to 31 years old (girls' mean age = 21.87, SD = 1.18; boys' mean age = 23.91, SD = 2.11). All of the tools were administered physically to the sample of pre-service teachers who were conveniently available during the pilot study. This was followed by a brief interview of eight randomly selected participants who were asked to comment on items that were included in each tool.

After analysis of each tool, five items were excluded from the EID scale, as they were perceived by the participants as contextually difficult to respond to (e.g., I keep mementos from the outdoors in my room, such as shells or rocks or feathers). Other items were modified accordingly to suit the study contexts. For example, an item that read "I really enjoy camping and hiking outdoors" was modified to read as "I really enjoy camping and walking outdoors". Overall internal consistency reliability of the EID scale was $\alpha = 0.927$, and the corrected item-total correlation was higher than 0.30 for all of the measured items, indicating that the scale items were highly reliable and measured the related constructs [58]. Likewise, the pilot study revealed that the internal consistency reliability for the 2-MEV was $\alpha = 0.618$. Some of the items were modified to suit the context of the study. For example, an item that read as "sitting at the edge of a pond watching dragonflies in flight is enjoyable" was modified to read as "sitting beside the water bodies watching flying insects is enjoyable", as the participants perceived the word dragonflies as uncommon to them. The same modifications were carried out on the EKT. Two items that related to ozone layer depletion, and another that related to recycling symbols were replaced, as more than 90% of the participants answered them correctly [59].

### 2.7. Data Analysis Procedure

Data cleaning was conducted to assess the suitability of the data to the analysis plan. Initially, missing values that exceeded 50% per construct were excluded from the dataset [17]. In total, 24 questionnaires were excluded (5 respondents in study 1 and 19 in study 2) from subsequent analysis. Additionally, the data were checked against outliers using standardized residual and Q-Q plots. Furthermore, the Kaiser–Meyer–Olkin (KMO) measure of sampling adequacy was computed, together with Bartlett's Test of Sphericity, in order to determine the underlying dimensions of the scales. Moreover, data were free

from multicollinearity, as all of the intercorrelations between variables were less than 5. Specifically, the maximum intercorrelation value was 0.409 between effectiveness and action-related knowledge.

Initially, exploratory factor analysis was conducted, in order to determine the number of underlying factors in the EID scale using eigenvalues greater than one. In order to confirm the proposed number of factors, parallel analysis was conducted using simulated eigenvalues. Two clearly demarcated factors were extracted (Table 1). Thereafter, a correlation analysis was conducted in response to the research questions. Given that the sample size from each of the selected TCs was less than 100 [60,61], each sample was analyzed independently using correlation analysis instead of combining different samples from each TC for the multiple regression analysis. Notably, this approach was used to control the study for pseudoreplication [62]. Specifically, Pearson's correlation coefficient was used to assess the relationships of variables in reasonably normally distributed data, whereas Spearman's rank correlation coefficient was used when the data were not normally distributed [51]. Additionally, the coefficient of determination ($R^2$) was computed alongside the values of the correlation coefficient, in order to determine the magnitude of the relationships between the compared variables [63].

**Table 1.** Factor structure of environmental identity scale.

| Environmental Identity Items (Variables of Interest) | Factors 1 | 2 | α |
|---|---|---|---|
| Behaving responsibly toward the environment and living a sustainable lifestyle are parts of my moral code. | 0.81 | | |
| Being a part of the ecosystem is an important part of who I am. | 0.79 | | |
| I think of myself as a part of nature, not separate from it. | 0.71 | | |
| Engaging in environmental behaviors is important to me. | 0.69 | | |
| In general, being part of the natural world is an important part of my self-image. | 0.67 | | |
| I have a lot in common with environmentalists as a group. | 0.56 | | |
| I feel that I have roots to a particular geographical location that had a significant impact on my development. | 0.49 | | |
| I feel that I have a lot in common with other species. | 0.43 | | 0.85 |
| Living near wildlife is important to me; I would not want to live in a city all the time. | | 0.76 | |
| I would feel that an important part of my life was missing if I was not able to get out and enjoy nature from time to time. | | 0.71 | |
| I have never seen a work of art that is as beautiful as a work of nature, like a sunset or a mountain view. | | 0.66 | |
| I really enjoy camping and walking outdoors. | | 0.59 | |
| My own interests usually seem to coincide with the position advocated by environmentalists. | | 0.53 | 0.82 |
| When I am upset or stressed, I can feel better by spending some time outdoors communing with nature. | | 0.50 | |
| I believe that some of today's social problems could be cured by returning to a more rural lifestyle in which people live in harmony with the environment. | | 0.42 | |
| If I had enough time or money, I would certainly devote some of it to working for environmental causes. | | 0.39 | |
| Kaiser–Meyer–Olkin measure of sampling adequacy | 0.920 | | |
| Approx. chi-square | 1816.323 | | |
| df | 120 | | |
| Sig. | 0.000 | | |
| Extraction Method: Principal Axis Factoring | | | |
| Rotation Method: Oblimin with Kaiser Normalization | | | |

Note: α = Internal consistency reliability.

In study 1, two correlation analyses were conducted, one before the study intervention, and another after the study intervention. In study 2, the relationships between measured variables were assessed using correlation analysis, which was conducted after the study intervention. In both studies, the analysis was controlled for gender, age, and SDR, considering these variables as potential moderators in measuring environmental constructs [33,64].

Internal consistency reliability was assessed using Cronbach's alpha coefficient and corrected item-total correlation [65]. For the EKT, test-retest reliability was computed. The psychometric properties of the scales are presented in Table 2. Seven items of the 2-MEV were deleted, and only thirteen (eight items for preservation and five items for utilization) items were used after reliability and validity assessments. Nonetheless, the details of the psychometric properties of the 2-MEV and the EKT are beyond the scope of this study.

**Table 2.** Reliability values of the data collection instruments.

| Variables | Pre-Test $\alpha$ Study 1 | Post-Test $\alpha$ Study 1 | Post-Test $\alpha$ Study 2 | Test-Retest Reliability (*r*) |
|---|---|---|---|---|
| 2-MEV | 0.653 | 0.720 | 0.735 | - |
| Identification with nature | 0.813 | 0.846 | 0.854 | - |
| Appreciation of nature | 0.787 | 0.818 | 0.799 | - |
| System knowledge | - | - | - | 0.864 |
| Action-related knowledge | - | - | - | 0.859 |
| Effectiveness knowledge | - | - | - | 0.939 |

## 3. Results of Study 1

### 3.1. Correlates of Environmental Identity before the Study Intervention

As indicated in Table 1, two correlated factors (domains) of environmental identity were determined before conducting the correlation analysis. Using oblique rotation, the extracted factors indicated that they were strongly correlated (*r* = 0.632), suggesting that they were measuring related underlying constructs. The first factor explained 38.7% of the variance, whereas the second factor explained 8.5% out of the total explained variance of 47.2% in the overall EID scale. Both of the extracted factors were highly reliable, and the intercorrelations (corrected item-total correlation) between items were adequate, ranging from 0.396 to 0.681 ($\alpha$ = 0.878) on the overall scale. The internal consistency reliability for each of the extracted factors is indicated in Table 1. For consistency of analysis, the first factor was named as identification with nature (IWN), whereas the second factor was named as appreciation of nature (AN). Three items did not load on either of the extracted factors, and were excluded from subsequent analyses. As such, only 16 items, 8 for each factor, were used in the analysis.

Preliminary analysis of normality tests using Kolmogorov–Smirnov revealed that domains of attitudes and identity were normally distributed (*p* > 0.050), whereas environmental knowledge dimensions were not. In addition, Z-scores were calculated by dividing the values of Kurtosis and skewness by their respective standard errors [66]. The obtained values confirmed the Kolmogorov–Smirnov values on the normality test. Therefore, the Pearson correlation coefficient was computed, in order to assess the relationships between environmental attitudes and domains of environmental identity. On the other hand, Spearman's rank correlation was calculated to assess the relationships between environmental knowledge dimensions and domains of environmental identity. The same procedure was used in both $TC_1$ and $TC_2$.

The study findings revealed that there was a positive correlation between preservation and identification with nature ($r(89) = [.47]$, $p < 0.001$) in $TC_1$. Likewise, effectiveness knowledge was positively correlated with identification with nature (IWN) among the participants of the same college ($r(89) = [.20]$, $p = 0.053$). Given the value of the coefficient of determination ($R^2 = 0.224$), the results suggest that about 22.4% of the variance of identification with nature could be explained by the relationship with the amount of

preservation attitude. The strength of the relationship between preservation and IWN is moderate, given that there is about 77.6% of the variance in IWN that is explained by other factors which are different from the preservation attitude (see Figure 2). On the other hand, the strength of the relationship between effectiveness knowledge (EfK) and IWN was weak [63].

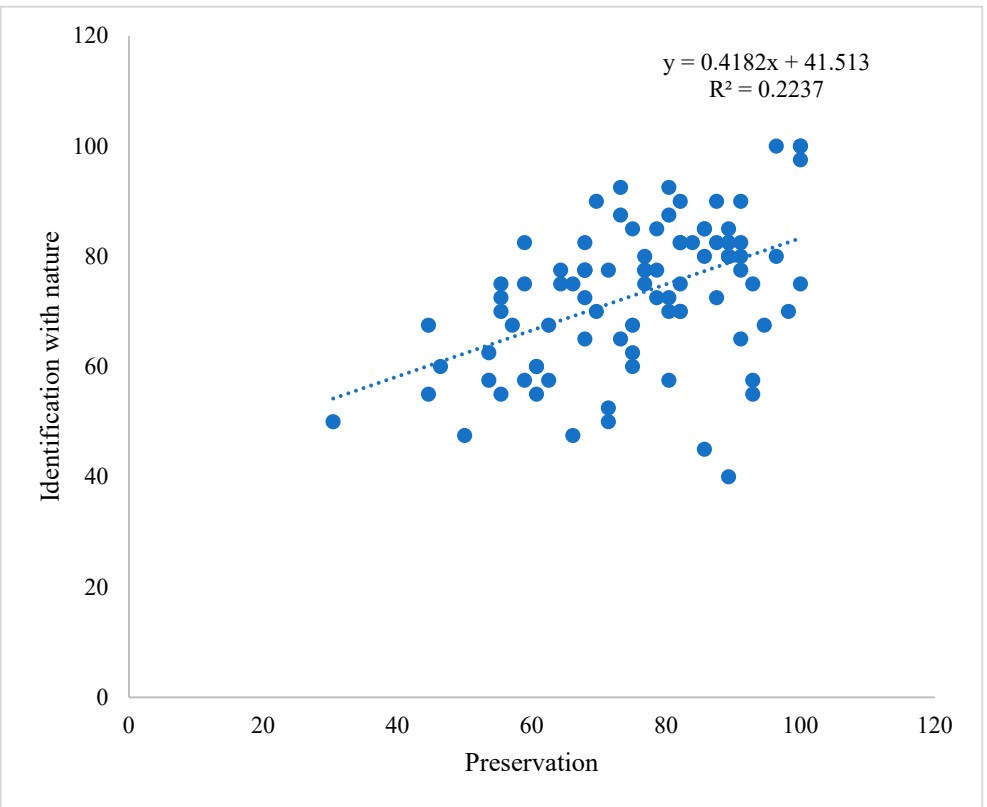

**Figure 2.** The relationship between preservation and identification with nature in $TC_1$. The dots in the graphs indicate that there were no serious outliers to affect the relationship whereas the dotted line indicate that the two variables were change together in the same direction.

On the contrary, utilization, system knowledge, and action-related knowledge were partially but not significantly correlated with IWN. Regarding the appreciation of nature, preservation was the only variable with a significant positive relationship with AN ($r(89)$ = [0.32], $p$= 0.002). However, the strength of the relationship was weak ($R^2$ = 0.101), indicating that only 10.1% of the variability in AN could be attributed to its relationship with preservation; that left nearly 90% of the variance as unexplained.

In $TC_2$, only two variables were reasonably normally distributed. Therefore, Spearman's rank correlation was used to assess the relationships between the measured variables. In particular, preservation and IWN were positively correlated ($r_s(67)$ = [0.46], $p < 0.001$), suggesting that a high score in preservation was related with a high score of AN. The strength of their relationship was moderate. Likewise, all of the knowledge dimensions were positively correlated with IWN despite the fact that the strength of the relationship was weak, with a declining pattern as follows: system knowledge ($r_s(67)$ = [0.39], $p = 0.001$), action-related knowledge ($r_s(67)$ = [0.31], $p = 0.009$), and effectiveness knowledge ($r_s(67)$ = [0.27], $p = 0.024$). Although utilization was partially negatively correlated with IWN, the correlation was not statistically significant ($r_s(67)$ = [−0.11], $p = 0.391$). Similar results were also evident for the relationships between attitudes and AN, as well as with the environmental knowledge dimensions. Specifically, preservation attitude was positively correlated with AN ($r_s(67)$ = [0.40], $p = 0.001$). Moreover, environmental knowledge dimensions were also positively correlated with AN in a decreasing pattern from system

knowledge ($r_s(67) = [0.36]$, $p = 0.003$), action-related knowledge ($r_s(67) = [0.27]$, $p = 0.026$), to effectiveness knowledge ($r_s(67) = [0.23]$, $p = 0.053$). On the contrary, utilization was partially negatively, but not significantly, correlated with AN ($r_s(67) = [-0.09]$, $p = 0.467$).

In general, preservation was the only variable with a moderate significant correlation with both domains of environmental identity, whereas other variables demonstrated either weak or nonsignificant correlations with environmental identity. An unexpected result was the positive correlation between social desirability responding (SDR) and utilization ($r(67) = [0.48]$, $p < 0.001$). As depicted in Figure 3, the coefficient of determination revealed that a considerable variability (23.2%) in utilization could be explained by its relationship with SDR. In TC$_1$, a similar result was observed in TC$_2$, although the strength of the relationship between utilization and SDR was weak ($r(89) = [0.37]$, $p < 0.001$), as reflected by the coefficient of determination ($R^2 = 0.1547$).

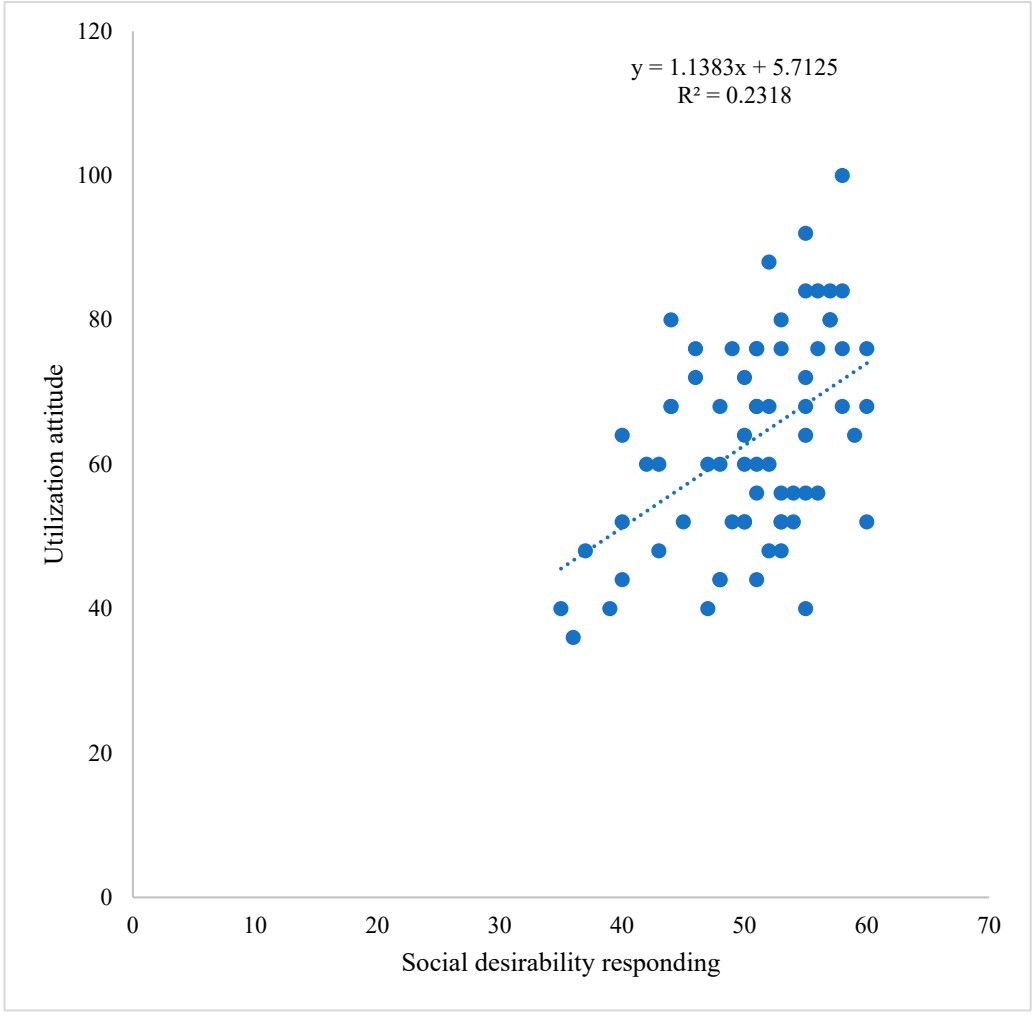

**Figure 3.** Correlation between SDR and utilization attitude in TC$_1$. The relationship direction as indicated by dots and straight line in Figure 3 is a positive relationship whereby SDR increases with an increase in the value of utilization attitudes.

### 3.2. Correlates of Environmental Identity after the Study Intervention

The same procedures that were used to assess the relationships between the measured variables before the study intervention were also employed after the study intervention. Spearman's rank correlation coefficients were computed to assess the relationships between the variables of interest. In TC$_1$, preservation ($r(89) = [0.35]$, $p = 0.001$) and effectiveness knowledge ($r(89) = [0.28]$, $p = 0.008$) were positively correlated with IWN, whereas the remaining variables were partially but not significantly correlated with IWN. Consequently,

the relationship pattern among the variables, before and after the study intervention, did not change. Thus, preservation and effectiveness knowledge appeared to be the only variables with a positive association with IWN in $TC_1$. Regarding the AN, preservation and effectiveness knowledge were positively correlated with AN, whereas other variables did not demonstrate a considerable relationship with AN in $TC_1$. Notably, effectiveness knowledge in the pre-intervention measurement was not significantly correlated with AN, as opposed to the post-intervention measurement.

In $TC_2$, Spearman's rank correlation coefficients were computed to assess the relationships between the variables. There was a positive correlation between preservation and IWN ($r_s(67) = [0.33]$, $p = 0.006$). In the same vein, all of the environmental knowledge dimensions were positively associated with IWN as follows: system knowledge ($r_s(67) = [0.39]$, $p = 0.001$), action-related knowledge ($r_s(67) = [0.30]$, $p = 0.011$), and effectiveness knowledge ($r_s(67) = [.29]$, $p = 0.017$). Consistently with pre-intervention measurement, utilization was not significantly correlated with IWN. Moreover, there was a positive correlation between preservation attitude with AN ($r_s(67) = [0.36]$, $p = 0.003$). Similarly, and consistently with previous measurements in the pre-test, all of the environmental dimensions were positively correlated with AN in the following pattern: system knowledge ($r_s(67) = [0.29]$, $p = 0.016$), action-related knowledge ($r_s(67) = [0.35]$, $p = 0.003$), and effectiveness knowledge ($r_s(67) = [0.27]$, $p = 0.028$). Again, utilization was not significantly correlated with AN in $TC_2$.

Finally, age was not correlated with any of the measured variables. However, SDR was positively correlated ($r = 0.37$, $p < 0.05$) with utilization in the pre-test of $TC_1$, but not with the post-test. Likewise, SDR and utilization were positively correlated in $TC_2$ during the pre- ($r = 0.39$, $p < 0.05$) and post-tests ($r = 0.45$, $p < 0.05$). There was no significant correlation between SDR and environmental identity. The strength of the correlation between SDR and utilization in the post-tests of subjects in $TC_2$ was moderate ($R^2 = 0.232$), suggesting that about 23.2% of the variability in the amount of utilization could be explained by the relationships with socially desirable responding, which was consistent with the pre-test results.

## 4. Discussion of Study 1

The main objective of study one was three-fold. Firstly, the study aimed at determining the construct validity of the EID scale; that is, determining the underlying domains in the EID scale that were correlated with environmental knowledge and attitudes. The study results indicated two correlated factors ($r = 0.632$) of the EID scale, identification with nature (IWN) and appreciation for nature (AN). This implied that the extracted factors shared 39.94% (when squared) of the variance in the EID scale. Thus, the environmental identity factors appeared to be measuring related constructs, and should be treated as domains of the same environmental construct. Internal consistency reliability indicated that the EID scale was highly reliable, both singly on each of the domains, and on the overall scale, and was consistent with previous research [10,13].

Secondly, the study established that preservation of nature was the only environmental construct that was positively correlated with both domains of environmental identity in both samples; at different measurement points, other variables demonstrated either weak or nonsignificant relationships with domains of environmental identity. This finding suggests that individuals with a high score on preservation attitude will as well tend to score high on both domains of environmental identity. In addition, the results suggest that promoting the extent to which people associate themselves with nature, and the degree of value they attach to nature, are largely dependent on protection and conservation attitudes. In addition, effectiveness knowledge appeared to be a significant variable next to preservation, and appeared to be more strengthened after the study intervention in $TC_1$, although the relationship was weak. In addition, environmental knowledge dimensions were all positively correlated with both domains of environmental identity, while utilization was not correlated with identity in any form. It should be noted, however, that the sample sizes

in $TC_1$ and $TC_2$ were not the same. It was not clear whether this would have affected the relationships between the measured variables. Study 2 provides more elaborate findings in response to this question.

Finally, age was not correlated with any of the measured variables. However, SDR was positively correlated ($r = 0.37$, $p < 0.05$) with utilization in the pre-test of $TC_1$, but not with the post-test. Likewise, SDR and utilization were positively correlated in $TC_2$ during the pre- ($r = 0.39$, $p < 0.05$) and post-tests ($r = 0.45$, $p < 0.05$). There was no significant correlation between SDR and environmental identity. While past research acknowledges that there are some possible correlations between SDR and environmental attitudes [33], there is a need to continue extending the measurement of SDR, not only in preservation, but also in other environmental constructs such as identity and utilization of nature.

## 5. Results of Study 2

*Correlates of Environmental Identity*

Spearman's rank correlation was used to assess the relationships between variables, given that only one variable (AN) was reasonably normally distributed in $TC_3$. The results revealed a positive correlation between IWN and other variables, except for utilization attitude. In order of their increasing relationships, the correlation values were as follows: system knowledge ($r_s(93) = [0.25]$, $p = 0.015$), action-related knowledge ($r_s(93) = [0.25]$, $p = 0.014$), effectiveness knowledge ($r_s(93) = [0.32]$, $p = 0.002$), and preservation of nature ($r_s(93) = [0.43]$, $p < 0.001$). Regarding AN with other variables, positive relationships were found between AN and other environmental variables, except for utilization and system knowledge. Specifically, the correlation levels were as follows: action-related knowledge ($r_s(93) = [0.25]$, $p = 0.013$), effectiveness knowledge ($r_s(93) = [0.28]$, $p = 0.006$), and preservation attitude ($r_s(93) = [0.32]$, $p = 0.002$).

In $TC_4$, environmental attitudes and both domains of identity were reasonably normally distributed, but the environmental knowledge dimensions were not. Thus, Pearson's correlation coefficient was used to assess the relationships between environmental attitudes and identity domains, whereas the Spearman's rank correlation was computed to assess the relationships between identity domains and environmental knowledge dimensions. Specifically, there was a positive relationship between IWN and preservation attitude ($r(76) = [0.53]$, $p < 0.001$). This was also true for utilization of nature, which was negatively correlated with IWN ($r(76) = [-0.25]$, $p = 0.031$). All of the environmental knowledge dimensions were not significantly correlated with IWN. Additionally, there was a positive correlation between preservation and AN ($r(76) = [0.45]$, $p < 0.001$), while all other variables did not indicate any significant correlation with AN. The findings suggest that about 27.6% and 19.9% of the amount of variability in IWN and AN, respectively, could be explained by their relationship with the amount of preservation attitude. The strength of their relationship was moderate, as indicated in Figure 4.

Age, as in study one, was neither correlated with environmental attitudes nor environmental identity domains in both $TC_3$ and $TC_4$. However, SDR was positively correlated with preservation ($r = 0.33$, $p = 0.001$, $R^2 = 0.1144$) in $TC_3$, but not correlated with utilization. Likewise, SDR was as well positively correlated with both IWN ($r = 0.23$, $p < 0.05$, $R^2 = 0.0523$) and AN ($r = 0.22$, $p < 0.05$, $R^2 = 0.0614$) in $TC_3$. Furthermore, SDR was positively correlated with preservation ($r = 0.33$, $p = 0.004$, $R^2 = 0.1062$), and with both IWN ($r = 0.33$, $p = 0.003$, $R^2 = 0.1087$) and AN ($r = 0.29$, $p = 0.009$, $R^2 = 0.0861$) in $TC_4$. Nonetheless, the relationship between SDR and other variables was weak, suggesting that more variance (nearly 90%) is explained by other factors that were apart from the socially desirable responses.

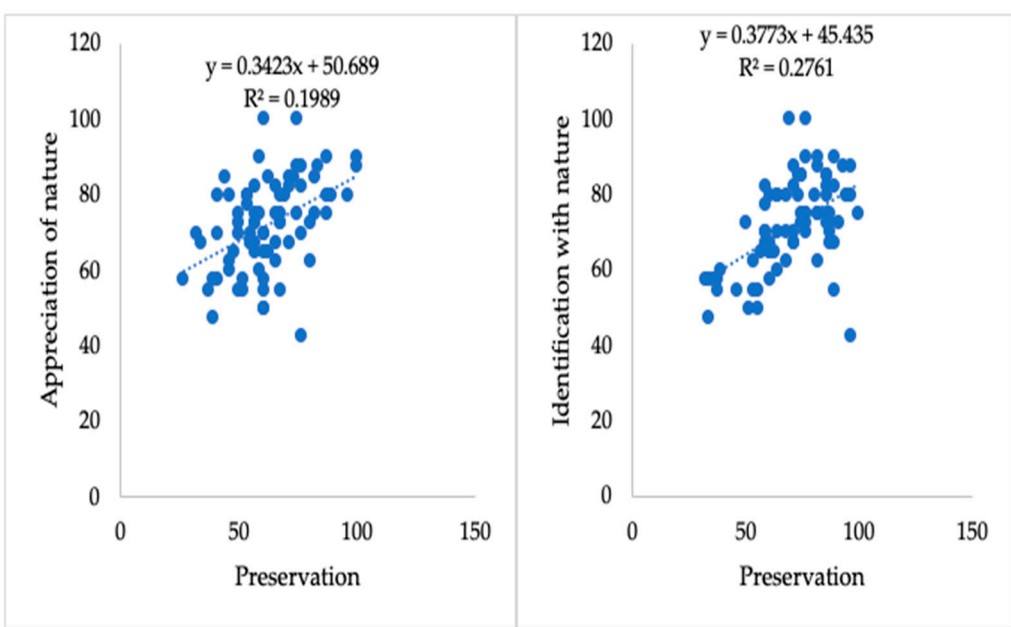

**Figure 4.** The relationships between preservation and domains of environmental identity in TC₄. The dots and straight lines in the figure suggest that the variables had proportional relationships implying that the values of preservation increased in the same pattern with an increase of the values domains of environmental identity. It should be noted, however, that correlation does not imply causation and one variable may not necessarily affect another variable.

## 6. Overall Discussion

### 6.1. An Overview of the Study Findings

Extant research has dedicated considerable effort to understanding the causal link between environmental identity and other environmental constructs, including personal values [67,68], connectedness with nature, and attitudes [14], which in turn predict pro-environmental behaviors [69]. Nevertheless, a lot remains unclear on the development of environmental identity (EI) in relation to environmental knowledge and attitudes. Thus, the current study reports on the findings of two studies, explaining the relationships between environmental knowledge dimensions and attitudes, and how they relate to specific domains of environmental identity. Specifically, environmental attitudes were measured and reflected using the 2-MEV model, whereas a Competence Model for Environmental Education was used as a measure of environmental knowledge dimensions.

Three consistent results were revealed in both studies. Firstly, environmental identity was found to be measured by two correlated domains, namely identification with nature and appreciation of nature; this was consistent with the results of some earlier studies [67,70]. As expected, this means that the EID scale can be considered to be a measure of two factors of environmental identity that are different, but still measuring related environmental constructs [10]. Secondly, both domains of EI were positively correlated with preservation in both studies, and in all of the measurement points. Therefore, it can be interpreted that people with preservation attitudes are more likely to associate themselves with nature than those who are not. In turn, this implies that people with high preservation and high environmental identity are more ecologically concerned and less environmentally destructive than those who are not [71]. Literally, the more an individual protects natural resources, the more likely he/she considers him/herself as part of nature.

Besides, the results suggest that environmental knowledge dimensions vary in their relations with identity across different populations. This calls for the need to employ context-specific approach in addressing issues that are related to environmental knowledge, particularly among pre-service biology teachers. In order of their contribution, effectiveness knowledge had a more nuanced relationship with identity compared with system and

action-related knowledge across studied samples. Partly, this could be explained by the fact that system and action-related knowledge tend to be distal determinants of environmental issues, while their formations strongly predict people's environmental decisions through their causal link with effectiveness knowledge [42].

Additionally, Goodwin and Leech [72] explain six factors that can cause weak to nonsignificant relationships between variables in correlational analysis. These factors include a lack of linearity between the compared variables, differences in shapes of two distributions, the amount of variability in data, the presence of outliers, measurement error, and sample characteristics.

In view of Goodwin and Leech's [72] assertion, however, we examined all possible reasons that could have caused weak to nonsignificant relationships of some of the measured variables, such as action-related knowledge, system knowledge, and more importantly, utilization attitude. Regarding linearity, the data displayed features of linearity, given that even ordinal data were converted to continuous data to meet the linearity feature. Furthermore, the participants in the sample did not vary considerably in terms of age and education level, given the sample characteristics. However, some of the samples displayed skewed distributions, particularly due to the ceiling effect, which seems to be a common experience in other earlier reported findings of environmental studies that measured environmental attitudes [73]. Given the possibility that social desirability can affect the strength of the relationship [51], there is a need to continue emphasizing the use of the social desirability scale in measuring environmental constructs, particularly for environmental attitudes.

Furthermore, the shape of a distribution may influence the value of the relationship. According to Goodwin and Leech [72], the variables are strongly correlated when the shape of the predictive (X) variable is equal to that of the outcome variable (Y). In our case, both distributions in each pair of the compared variables had the same shape. Therefore, this eliminates the possibility of this factor being considered as one of the limiting factors for the relationships between the variables.

Notably, the socio-cultural and economic backgrounds of participants were not examined in breadth. Remarkably, this could be one of the potential factors to be explored in future research. Moreover, utilization was only correlated with identity in only one of the measurement points. This needs to be interpreted with caution, as the relationship of utilization with identity may have been influenced by sample overlap, rather than on the actual relationships with other variables [74]. Given the possibility of a remarkable socially desirable response observed with utilization, further investigation may be useful to provide more robust findings in using a larger sample.

On the other hand, Glenberg and Andrzejewski [74] caution that combining groups from different samples tends to distort correlational analyses. In turn, this may also lead to pseudoreplicated findings [62]. Although pseudoreplication appears to be more common in biological studies, more than half of the published studies on experiments in other fields are pseudoreplicates [75]. As such, our analyses did not combine samples from different studies in an effort to control for pseudoreplication. In so doing, we may have affected statistical power that would have been more sensitive to the relationships between the measured variables.

Measurement errors, such as fatigue, scoring errors, few numbers of items, guessing, and ambiguities of the items, can also lead to limited correlations between the measured variables [72]. Nonetheless, the numbers of items in our studies were adequate, as supported by other previous studies that measured environmental knowledge [76], attitudes [38], and identity [10]. Again, few numbers of items cannot be a reason for the observed weak relationships of some variables in our studies, given the efforts that were invested in validating the tools. In addition, all of the study tools were piloted before the actual studies in order to improve their quality as explained in the previous section. As such, we did not expect the ambiguity of items to be one of the limiting factors. Equally, all of the items were scored in the same ways as other previous studies for the purpose

of maintaining consistency. Therefore, we do not expect a scoring error to be among the limiting factors, as the validity and reliability of the items were considered to be within the broad array of acceptable values. Additionally, problematic items were removed, as one of the strategies to improve the internal validity of our studies and of the results [50]. Ultimately, all of the tools were administered to the participants at different times, mainly during normal learning hours. We believe that this measure outweighed the problem of fatigue as a limiting factor for weak correlations between some of the variables.

### 6.2. The Effect of Age and Social Desirability on the Relationships of the Variables

Past research has revealed that the effect of social desirability responding is one of the potential biases in environmental psychology research, having a greater impact on preservation, while utilization largely remains uncontaminated [33]. Conversely, mixed findings in which social desirability did not show a moderating effect on the relationships between attitudes and other environmental constructs have also been reported in previous research [77]. Likewise, the age of respondents has also been reported as an important determinant of environmental identity and other environmental constructs [78]. Given the usefulness of assessing the effects of age and SDR on the relationships between independent and outcome variables in self-reported scales, we administered the SDR scale, along with other instruments. The results reported in this study revealed that there was no significant effect of age on the relationships between the measured variables.

However, notable positive correlations between SDR and other variables were observed. These findings suggest that the measurement of SDR should not be overlooked. The high correlations, for example, observed between SDR and utilization among participants in $TC_2$ necessitates the need for a more controlled administration of self-reported tools, and a cautious interpretation of the findings from self-reported responses.

Moreover, by using two studies, Milfont [79] found that SDR had a weak direct effect on EA, and had no moderating effect on the relationships between EA and ecological actions. Likewise, a meta-analysis of 29 previously published papers that was conducted by Vesely and Klöckner [80] revealed that the pooled correlations with SDR were small, ranging between 0.06 to 0.11, and 0.08 to 13, after correcting for measurement of error attenuation. Nonetheless, the authors recommended the need to continue assessing for the effects of SDR in future research, despite its marginal impact on other reported findings. The potentiality of assessing the effects of SDR is due to cross-cultural variations, where self-reported scales are administered that may result in potential biases in responding to the scale items [81]. Ultimately, some reported potential effects of impression management suggest a possible effect of SDR on environmental questions [82]. Consequently, both scholars and lay people still possess the strong belief that pro-environmental actions are influenced by their social desirability [83].

### 6.3. Practical and Theoretical Implications of the Results

The results that were reported in this study provide remarkable contributions to the usefulness of EA, knowledge, and identity in fostering environmental stewardship. Previous research has consistently reported on the efficacy of environmental identity in mediating ecological behaviors [19,68,84]. Similarly, environmental identity has been described to be a potential construct in shaping people toward energy transition [85]. Furthermore, environmental identity is not only beneficial to the environment per se, but past research has also demonstrated EI to be an important determinant of personal calmness, increased attentiveness, reduced stress, and mental health [4]. Therefore, our results provide insights that suggest that targeting specific environmental attitudes, such as enhancing preservation, works better to strengthen environmental identity. Partly, the results propose a need to promote knowledge about individual behaviors that are most effective in attaining effective protection and conservation of environmental resources. For policymakers, the results provide fundamental information that calls upon the need to integrate preservation movements, and to create awareness among individuals on their

actions that are environmentally impactful. In turn, policymakers may integrate these environmental aspects, in order to promote environmental sustainability.

For researchers, the findings supported the unidimensional latent construct of the EID scale, consistent with the original proposition [13], and of the revised version of the scale [17]. Likewise, these results corroborate with the unidimensional latent construct of the EID scale that was reported by Moreira et al. [10]. Thus, our results support the argument provided by Moreira and colleagues, stating that previous research that reported multidimensional solutions of the EID scale items [16], "may have done so, not because environmental identity is a multidimensional construct, but because the conceptual breadth of the construct results in diverse item content" (p. 7). In summary, our results provide additional evidence that suggests that the EID scale is cross-culturally reliable and valid for measuring environmental identity, provided that the items are modified to suit context-specific applications.

### 6.4. Limitations

Despite the remarkable and useful results reported herein, it is worth noting some important considerations for future research. In particular, the results reported in this article were from two studies that were coherently conducted among pre-service biology teachers. While the Tanzanian community is composed of heterogenous cultural contexts, the studies did not manage to include every single useful construct that would have helped to understand the relationships between the measured variables on a broader scale. Thus, future research may benefit from extending the sample to other target individuals, including teachers in schools, students in universities, or to any other category, in order to understand how they shape the development of environmental identity. Additionally, environmental identity is associated with other variables that are beyond those measured in our studies. For instance, personal meaning, emotional well-being, and rural-urban residency are among the potential determinants [86]. Still, other causal links between personal values and environmental self-identity have been reported as important antecedent links between identity and ecological actions [87]. However, our primary purpose was not to understand the comprehensive determinants of environmental identity. Thus, our target was to explain how specific attitudes and knowledge related to specific domains of environmental identity, and this can be used as a fundamental aspect for understanding the interrelations between other variables in future research.

Given the literature review conducted by Simms [23] on the operational definitions of environmental identity, our results focused on only one dimension of conservation movements' identities, while other domains of Meadian identity theories and Eriksonian identity theories remained comparatively unreported in our results. Therefore, more research is needed to understand other dimensions of environmental identity.

Our studies were conducted in four different TCs. Therefore, combining any of these TCs in any form would have been translated as a pseudoreplication of the study [52,53]. According to Lazic [62], there are three important requirements that should be met for a genuine replicated experimental study. Specifically, the proposed requirements include the following: firstly, the study subjects should be independent, and randomly assigned to the study groups (control and experimental groups); secondly, ecological units (variables) must be independently replicated for valid inference; thirdly, experimental units (subjects) should not influence each other. Our studies employed intact classes of pre-service biology teachers, in which randomization of the subjects to the study groups is sometimes translated as being unethical in educational studies [50]. Therefore, this posed a methodological limitation that may have affected statistical power in the analyses. Thus, the interpretation of the findings needs to be conducted with caution. Nonetheless, our study findings provide fundamental insights to be used in more controlled future studies, particularly through randomized controlled trials [88].

## 7. Conclusions

Given the study results, the theory of ecological attitudes (2-MEV) provides a novel contribution in understanding domains of environmental identity. Particularly, people who tend to score high on the preservation domain are likely to score high on both domains of environmental identities. Thus, shaping the extent to which people define themselves in relation to nature is largely dependent on changing their attitudes towards the conservation and protection of nature. Moreover, the attitude domains of preservation and utilization interact differently with environmental identity. In particular, the relationship between preservation and environmental identity was moderate and consistent in both studies whereas utilization seemed to be partially related to identity. In the same way, the interactions of environmental knowledge dimensions with identity varied considerably. Unexpectedly, system and action-related knowledge were partially correlated with domains of identity. We did not find an adequate explanation for this observation, given that there were some notable correlations between SDR and some of the measured variables. This has been described as one of the limitations for variable relationships when young people are engaged in self-reported questionnaires [47]. Likewise, we did not find support for a demographic variable of age being among the moderating factors, given its limited correlations with the measured variables. Ultimately, the role of the Competence Model for Environmental education in understanding environmental identity was empirically supported as a crucial component in studying pre-service biology teachers' environmental identity.

Finally, we propose a more systematic hypothesis to be empirically tested, using a different sample with more heterogeneous features, and a randomized controlled trial study. Given the difficulties in conducting randomized controlled trials in education, it would be more appealing to test the predictive power of environmental attitudes and knowledge over environmental identity in multicultural contexts.

**Author Contributions:** Conceptualization, J.P.N.; methodology, J.P.N.; validation, all authors; formal analysis, J.P.N.; investigation, J.P.N.; supervision, C.M.A. and F.N.; data curation, all authors; writing—original draft preparation, all authors; writing—review and editing, all authors; funding acquisition, J.P.N. All authors have read and agreed to the published version of the manuscript.

**Funding:** This research was funded by the African Centre of Excellence for Innovative Teaching and Learning Mathematics and Science (ACEITLMS), grant number ACE II(P151847).

**Institutional Review Board Statement:** This study was conducted according to the guidelines of the Declaration of Helsinki, and approved by the Institutional Review Board of the College of Education of the University of Rwanda (Protocol # 01/P-CE/745/ENg/2019 approved 23 September 2019).

**Informed Consent Statement:** Informed consent was obtained from all subjects involved in the study.

**Data Availability Statement:** Data can be obtained by contacting the corresponding author after publication.

**Acknowledgments:** Thanks to all teacher colleges' administrations for their commitment, and for accepting the idea of this research to be conducted on their respective campuses. Furthermore, all pre-service biology teachers in their respective teacher colleges, who participated in both studies, are highly appreciated for their valuable contributions to making this research feasible. Finally, we would like to extend our gratitude to the three reviewers for their valuable comments that significantly improved the quality of the original as well as the revised submitted manuscript.

**Conflicts of Interest:** The authors declare no conflict of interest.

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
