# Peer review of "An Empirical Investigation of Environmental Knowledge and Attitudes as the Correlates of Environmental Identity among Pre-Service Biology Teachers in Tanzania"

_sustainability, doi:10.3390/su15010669_

Round 1
Reviewer 1 Report
Please see the attached file.

Author Response
Response to Reviewer 1 Comments
Point 1: This manuscript contains interesting original data. However, this manuscript contains critical errors in the design of the study. Combining different groups (colleges) into one group is pseudoreplication. Pseudoreplication is defined as a gross statistical error. Pseudoreplication can lead to artificially inflated degrees of freedom, giving the illusion of having a more powerful test than the data support (Hurlbert 1984, 2009). This manuscript contains a large number of typographical errors. Degrees of freedom (F(df1,df2) were incorrectly calculated in the ANOVA model.
Response 1: First, I I wish to appreciate a very critical comment raised by the reviewer and in particular on the issue of pseudoreplication. I have gone through many articles to read about in and I have enriched the revised manuscript. Specifically, re-analysis has been conducted for all the study findings by treating each sample independently. The methodology section has been updated to accommodate the changes in the analysis. The discussion section, limitation, and conclusion sections all have been revised in accordance with the proposed changes.
Point 2: Participants, study 1. According to the design of this study 1, total sample size (N), N = 160, where females, n1 = 86 and males, n2 = 74. Study participants were selected from two teacher colleges (TC1, N1 = 91 and TC2, N2 = 69) from two different regions in Tanzania.
Thus, there are two major obstacles to grouping the two colleges into one group: the sample size is N = 160. First, two students from the same college are more similar to each other than two students from different colleges. Because two students from the same college have a common information space, and two students from different colleges do not have such a common information space. This design study is called sacrificial pseudoreplication. The use of sacrificial pseudoreplication can lead to artificially inflated degrees of freedom, giving the illusion of having a more powerful test than the data support (Hurlbert, S.H. The ancient black art and transdisciplinary extent of pseudoreplication. J. Comp. Psychol. 2009, 123, 434–443). Secondly, the gender difference in the perception of environmental identity is a working hypothesis in the study of human interaction with the environment. I believe this will be the strength of this study. If the two colleges maintain a ratio of 53.8% and 46% for women and men, respectively, then the minimum sample size would be n > 30.
In this context, Pearson's parametric correlation or Spearman's nonparametric correlation tests can be used instead of multiple regression analysis. To use multiple regression analysis, the current sample is very small.
According to Boldina, I.; Beninger, P.G. Strengthening statistical usage in marine ecology: Linear regression. J. Exp. Mar. Biol. Ecol. 2016, 474, 81–91, in order to determine Pearson’s correlation, five Gauss–Markov assumptions should be met: (1) linear in form (residuals vs. predicted) i.e., visual examination of the residuals plot, (2) no correlation between residuals and independent variables,(3) absence of autocorrelation in residuals, (4) homoscedasticity, and (5) normality in residual distribution. If one of the five assumptions was not met, then the Spearman correlation should be used.
Response 2: As noted, the changes regarding pseudoreplication can be traced in the revised manuscript from page 7 on and throughout the revised manuscript. For data that are normal, the correlation analysis has been computed using appropriate technique. Similarly for data that were non-normal, correlation analysis has been computed using the Spearman rank correlation technique. The recommended references have been read extensively and some of them have been cited in the revised manuscript to improve the quality of the revised manuscript.
The minimum sample size after treating each sample separately is adequate (>30) and the required ratios for female and male participants has been considered.
Point 3: Participants, study 2. The design of this study 2 has the same problems as study 1.
The results section should contain the raw data in the plots. Tables 3-5 should be deleted as there is no original date in these tables. Figures 1 and 2 should be deleted as the data should be presented in separated small groups (see above). Effect size and exact p-value should be presented in the results section.
Lines 353-356 and Table 3. It was written: “The ANOVA statistic was significant (F(5, 159)=10.582, p<.001)”.If F(5, 159) than there six independent group (df1=G-1, df1 = 5). According to Table 3 three are five dependent variables (PRE UTL SK ARK EfK) and 1
independent (IWN) in top panel and five dependent variables (PRE UTL SK ARK EfK) and 1 independent (AN) in bottom panel. It was written "df2=159", this is not correct, df2 = N-1-G, 139-6=133. If five measurements are taken on one object (experimental setup) and then processed as five independent groups, then this is a simple pseudoreplication (Hurlbert 1984, 2009).
Lines 444-445 and Table 4. It was written: “The overall model was signifi-444 cant (F(5, 167)=16.341, p<.001)” Table 4 contains the same problem as table 3.
Response 3: After revising the manuscript as recommended, Tables 3-5 have been removed given the updated analysis. Degree of freedom and p-values have been computed and presented in the revised manuscript.
On a separate note, the title of the manuscript has been changed given the focus of the revised analysis.
Once agin, thank you very much for your constructive inputs.
Reviewer 2 Report
The article is based on an interesting idea and presents an appropriate topic. The following is a list of comments:
1.The article should be modified to ensure that it has the structure of an academic research article, which would generally follow the typical framework of (Introduction) - (Literature review)- (Method) - (Results and discussion) - (Conclusion).
2) Introduction. The general discussion in the introduction should be limited. There is no need to repeat well-known problems. The introduction should include the problem context.
3) Literature review -There should be more emphasis on the rationale of the research and reference can be made here to solutions proposed by e.g.: Chomać-Pierzecka, E.; Sobczak, A.; Urbańczyk, E. RES Market Development and Public Awareness of the Economic and Environmental Dimension of the Energy Transformation in Poland and Lithuania. Energies 2022, 15, 5461. https://doi.org/10.3390/en15155461.
4) Methodology. The development of an appropriate methodology is crucial for scientific work. It is worth using a diagram or drawing illustrating this area in the methodology.
5) Discussion. This segment should be enriched with additional results and comparisons with other studies in the field. Furthermore, the benefits of the proposed model are not clear enough and should be described more.
6) Conclusion. The last chapter should be expanded and the practical implications of your research should be elaborated in more detail. It would make sense if you discussed the results, suggesting how they are relevant for further development.
Author Response
Response to Reviewer 2 Comments
Point 1: The article should be modified to ensure that it has the structure of an academic research article, which would generally follow the typical framework of (Introduction) - (Literature review)- (Method) - (Results and discussion) - (Conclusion).
Response to point 1: I appreciate the comment provided by the reviewer. However, the sections included in the manuscript followed the sample provided by the journal in which the literature review is fused with the introduction section. Nevertheless, the revised manuscript has been improved.
Point 2: The general discussion in the introduction should be limited. There is no need to repeat well-known problems. The introduction should include the problem context.
Response to point 2: revision has been made to accommodate this useful comment. In particular, issues such as specific SDGs that were mentioned have been removed from the introduction section to keep more focused on the problem context.
Point 3: Literature review -There should be more emphasis on the rationale of the research and reference can be made here to solutions proposed by e.g.: Chomać-Pierzecka, E.; Sobczak, A.; Urbańczyk, E. RES Market Development and Public Awareness of the Economic and Environmental Dimension of the Energy Transformation in Poland and Lithuania. Energies 2022, 15, 5461. https://doi.org/10.3390/en15155461.
Response to point 3: Revision has been made and the proposed reference has been consulted and cited. However, more focus has been given to the implication of the study findings as we think they speak more of the solutions and usefulness of the study than other sections.
Point 4: Methodology. The development of an appropriate methodology is crucial for scientific work. It is worth using a diagram or drawing illustrating this area in the methodology.
Response to point 4: Figure 1 in the revised section provides an elaborate figure as recommended by the reviewer
Point 5: Discussion. This segment should be enriched with additional results and comparisons with other studies in the field. Furthermore, the benefits of the proposed model are not clear enough and should be described more.
Response to point 5: In the revised version, some new analysis has been conducted that has also reshaped both the discussion section and partly the conclusion of the study. Therefore, we feel that we might have addressed this comment by adjusting the discussion to the changes made in the analysis. See details in the revised manuscript.
Point 6: Conclusion. The last chapter should be expanded and the practical implications of your research should be elaborated in more detail. It would make sense if you discussed the results, suggesting how they are relevant for further development.
Response to point 6: This section of the revised manuscript has been updated. Specifically, the last section has even further proposed the hypothesis that can be developmentally investigated in future research. Limitations of the study, implications of the study findings as well as the conclusion sections all have been updated accordingly.
Reviewer 3 Report
Dear authors,
Thank you for the opportunity to review the paper.
The manuscript entitled "The Efficacy of Environmental Attitudes and Knowledge in Predicting Environmental Identity of Pre-service Biology Teachers" is exciting and up-to-date.
The paper presented deals with a fascinating and important topic from a sustainability point of view. It needs to be treated as a critical voice in the discussion, the proposed holistic approach is particularly valuable to the raised issue.
The manuscript presents the results of interesting studies that were carried out in accordance with the preparation methodology. Figures and tables are properly prepared. The research was done in detail, following all methodological procedures.
Overall the presentation is reasonably good, but it might still require some work.
· The purpose of the manuscript is not clearly stated in the Abstract.
In conclusion, congratulations on your exciting manuscript.
Author Response
Response to Reviewer 3 Comments
Point 1: The purpose of the manuscript is not clearly stated in the Abstract.
Response to point 1: The purpose of the manuscript has now been clearly stated in the manuscript. Other sections of the manuscript have also been improved in the revised manuscript.
Round 2
Reviewer 1 Report
In general
The authors have made significant progress in understanding pseudoreplication. However, the materials and methods section does not provide important information about the sample size of males and females at each college. The results section should contain figures that visually represent the raw data. The results section should not contain method explanations. The following sections: "Abstract", "Materials and Methods " and "Results" should be rewritten.
In specific
Lines 20-22. The abstract does not contain a clear description of the study design.
It was written “Study one (N=160) was an intervention with pre and post-measurements involving indoor and outdoor environmental programs among pre-service biology teachers in two TCs”.
If I understand correctly, the first study re-measured "before" and "after" in two separate colleges. In this context, the total sample size (N) was N = 160. Thus, N contains the sample size (n), n1 = 91, males from two colleges and n2 = 69, females from two colleges. However, it is not known how many men and women were in each college. This information is the key to understanding the results of the study.
Lines 22-23. It was written “Study two (N=173) was conducted in two other TCs with post-test only”. In this context, N = 173 includes n1 = 78, males from two colleges and n2 = 95, females from two colleges. However, it is not known how many men and women were in each college. This information is the key to understanding the results of the study.
Line 26. It was written “Multiple correlation analyses revealed that…”
Authors used either Pearson’s product–moment correlation or Spearman’s rank-order correlation. In this context, “Multiple correlation analyses” is not correct definition. In this context, "Multiple correlation analyses" is not the correct definition.
Line 205. It was written “TC1 (n=91; males=53.8%) and TC2 (n=69; females=63.8%)” where TC is a teacher college. However, 91 + 69 = 160. It's not clear. Because N = 160 is the total sample size for Study 1. However, it is not known how many males and females are in each college. This information is the key to understanding the results of the study.
Lines 210-111 It was written “…TC3 (n=95; females=53.7%) and TC4 (n=78; males=56.4%)…” It is not known how many males and females are in each college. This information is the key to understanding the results of the study.
The reference [54] contains theoretical example when instead paired t-test unpaired t-test was used. At the same time, the degrees of freedom were artificially overestimated, which led to a Type II error (i.e., to the acceptance of a false null hypothesis). In a previous version of the current manuscript, sacrificial pseudoreplication took place (see Hurlbert 2009). Sacrificial pseudo-replication may result in a Type I error (i.e., rejecting the true null hypothesis).
Pseudoreplication definitions must be correct and precise. Pseudoreplications arise due to errors in study design and/or due to incorrect application of a statistical test.
Figure 1 does not include information about males and females in each college. This information is the key to understanding the results of the study.
Lines 368. It was written “…(r(89)=[.47], p<.001) in TC1…”
Line 370. It was written “…(r(89)=[.20], p= .053…)”
If I understand correctly, (89) is sampling size from pooling two colleges, males.
The authors should explain how the sample size n = 89 was obtained?
Interpretation of the correlation coefficient (|?|) in terms “Very Weak”, “Weak”, “Moderate”, “Strong”, and “Very Strong”. For parametric correlation, the effect size needs to be calculated. Sometimes a low p-value can be associated with a negligible effect size, since the p-value also depends on the sample size.
Lines 380-393. It was written “r(67)”.
The authors should explain how the sample size n = 67 was obtained?
Lines 465-470. It was written “r(93)”
The authors should explain how the sample size n = 67 was obtained?
Lines 476-478. It was written “r(76)”
The authors should explain how the sample size n = 67 was obtained?
Author Response
Response to Reviewer Comments
Point 1: The authors have made significant progress in understanding pseudoreplication. However, the materials and methods section does not provide important information about the sample size of males and females at each college. The results section should contain figures that visually represent the raw data. The results section should not contain method explanations. The following sections: "Abstract", "Materials and Methods " and "Results" should be rewritten.
Response to point 1: We have made changes in the pointed areas. Specifically, the number of participants in each college has been clearly stated in Figure one using both actual figures and related percentages.
Figures that represent raw data have been included in the revised manuscript including the coefficient of determination to provide a more robust interpretation of the results (see Figures 2 to 4).
Point 2: Lines 20-22. The abstract does not contain a clear description of the study design.
Response to point 2: Line 19 of the revised manuscript provides a snapshot of the design of the study in the abstract.
Point 3: It was written, “Study one (N=160) was an intervention with pre and post-measurements involving indoor and outdoor environmental programs among pre-service biology teachers in two TCs”.
If I understand correctly, the first study re-measured "before" and "after" in two separate colleges. In this context, the total sample size (N) was N = 160. Thus, N contains the sample size (n), n1 = 91, males from two colleges, and n2 = 69, females from two colleges. However, it is not known how many men and women were in each college. This information is the key to understanding the results of the study.
Response to point 3: In the revised version of the manuscript, the information about the participants in the study has been clearly stated supported by a Figure that depicts the number of each category of participants in both colleges.
It should also be noted that, despite the fact that data were collected in different colleges, each college has been treated separately in the analysis and interpretation of the study findings to avoid pseudoreplication as recommended by the reviewer in the first round of the review process.
Point 4: Lines 22-23. It was written, “Study two (N=173) was conducted in two other TCs with post-test only”. In this context, N = 173 includes n1 = 78, males from two colleges, and n2 = 95, females from two colleges. However, it is not known how many men and women were in each college. This information is the key to understanding the results of the study.
Response to point 4: Again, this information has been updated both in the text of the revised manuscript and in the graphical presentation in Figure 1.
Point 5: Authors used either Pearson’s product-moment correlation or Spearman’s rank-order correlation. In this context, “Multiple correlation analyses” is not the correct definition.
Response to point 5: This is also a very constructive comment like others. Therefore, revision has been made to accommodate a consistent use of appropriate terms in different sections of the manuscript.
Point 6: Line 205. It was written, “TC1 (n=91; males=53.8%) and TC2 (n=69; females=63.8%)” where TC is a teacher college. However, 91 + 69 = 160. It's not clear. Because N = 160 is the total sample size for Study 1. However, it is not known how many males and females are in each college. This information is the key to understanding the results of the study.
Response to point 6: In the revised manuscript, the information about the number of participants and category in each case has been clarified. Notably, all the analyses conducted were computed in consideration of each college as an independent sample.
Point 7: Lines 210-111 It was written “…TC3 (n=95; females=53.7%) and TC4 (n=78; males=56.4%” It is not known how many males and females are in each college. This information is the key to understanding the results of the study.
Response to point 7: The same revision was done in all sections with queries on gender clarification to provide a good pictorial representation of the study findings. Figure 1 presents well summarized information.
Point 8: The reference [54] contains a theoretical example when instead paired t-test unpaired t-test was used. At the same time, the degrees of freedom were artificially overestimated, which led to a Type II error (i.e., to the acceptance of a false null hypothesis). In a previous version of the current manuscript, sacrificial pseudoreplication took place (see Hurlbert 2009). Sacrificial pseudo-replication may result in a Type I error (i.e., rejecting the true null hypothesis).
Pseudoreplication definitions must be correct and precise. Pseudoreplications arise due to errors in study design and/or due to incorrect application of a statistical test.
Response to point 8: We have tried to comply with the suggestions provided by the reviewer including, in particular, re-analyzing the data from each college separately.
We believe that the issue of sacrificial pseudoreplication has been addressed after treating each sample separately in the analysis. However, we are still open to accepting suggestions where the reviewer may still find it useful to improve the manuscript for publication. We are very keen and thankful for the comments raised by the reviewer, particularly on pseudoreplication and on other areas.
Point 9: Figure 1 does not include information about males and females in each college. This information is the key to understanding the results of the study.
Response to point 9: We have revised the Figure by including specific important information as recommended by the reviewer.
Point 10: Lines 368. It was written, “…(r(89)=[.47], p<.001) in TC1…”
Line 370. It was written “…(r(89)=[.20], p= .053…)”
If I understand correctly, (89) is sampling size from pooling two colleges, males.
The authors should explain how the sample size n = 89 was obtained?
Response to point 10: 89 is a degree of freedom (n-2). The sample size is n=91 in TC1 it was obtained from one college (TC1) as an intact class as explained in line 205. Lines 204 to 210 specifically indicates the number of females and males in each college. Besides, detailed information is provided in Figure 1.
This also applies to TC2 to TC4. As stated above, our revised analysis did not combine any of the colleges and treat them as a unified sample. Instead, two colleges are in each study but the analysis treats them as an independent sample and did not combine them during the analysis.
Point 11: Interpretation of the correlation coefficient (|?|) in terms “Very Weak”, “Weak”, “Moderate”, “Strong”, and “Very Strong”. For parametric correlation, the effect size needs to be calculated. Sometimes a low p-value can be associated with a negligible effect size, since the p-value also depends on the sample size.
Response to point 11: The revised manuscript contains Figures (Figure 2 to 4) that visually indicate the correlation coefficient, equations, and coefficient of determination (R2) to simplify the interpretation of the correlation coefficient.
Point 12: Lines 380-393. It was written “r(67)”.
The authors should explain how the sample size n = 67 was obtained?
Response to point 12: As noted in the previous section, the sample was obtained using a single-stage cluster random sampling procedure. In that regard, colleges that offer science diploma courses were identified (10 TCs) out of which four were simple randomly selected and used to serve the purpose of this study (see lines 221-224 for details).
Point 13: Lines 465-470. It was written “r(93)”
The authors should explain how the sample size n = 67 was obtained.
Response to point 13: r(93) contains information on the correlation coefficient and degree of freedom (n-2). In TC3, the sample size was 95 obtained using the same procedure as stated above and in the methodology section. Therefore, 93 is a degree of freedom (n-2). Should there be some useful recommendations on this, we will appreciate for improving the quality of our manuscript.
Point 14: Lines 476-478. It was written “r(76)”
The authors should explain how the sample size n = 67 was obtained?
Response to point 14: As stated above, 76 is a degree of freedom in TC4 that had a sample size (n) of 78. 67 is also a degree of freedom in TC1 (n=69). Therefore, these figures were obtained from the sample size of each college.
In general, the reviewer raised very important issues that required clarification. We are very thankful to the reviewer for such very constructive comments that have really made significant contributions to improving the quality of this manuscript.
Round 3
Reviewer 1 Report
Accept in present form.